# The Hot QTL Locations for Potassium, Calcium, and Magnesium Nutrition and Agronomic Traits at Seedling and Maturity Stages of Wheat under Different Potassium Treatments

**DOI:** 10.3390/genes10080607

**Published:** 2019-08-12

**Authors:** Xing Shen, Yapei Yuan, Han Zhang, Ying Guo, Yan Zhao, Sishen Li, Fanmei Kong

**Affiliations:** State Key Laboratory of Crop Biology/National Engineering Laboratory for Efficient Utilization of Soil and Fertilizer Resources, Shandong Agricultural University, Tai’an 271018, China

**Keywords:** wheat, potassium, calcium, magnesium, recombinant inbred lines, quantitative trait locus, morphological trait

## Abstract

Potassium (K) is one of the most important mineral nutrients for wheat. In this study, the effects of low K (LK) treatments and the quantitative trait loci (QTLs) for K, calcium (Ca), and magnesium (Mg) use efficiency traits, both at the seedling and maturity stages of wheat, were investigated. The set of “Tainong 18 × Linmai 6” recombinant inbred lines (RILs) were used to identify the QTLs under different K treatments using hydroponic culture and field trials. The majority of K concentrations and content-related traits at seedling and maturity stages decreased with reduced K supply, but the K use efficiency-related traits increased. In contrast, with reduced K supply, the contents of Ca and Mg increased, while the Ca and Mg use efficiency decreased. A total of 217 QTLs for seedling traits and 89 QTLs for adult traits were detected. Four relatively high-frequency QTLs (RHF-QTLs) and 18 QTL clusters (colocation of QTLs for more than two traits) were detected. Eight clusters were detected for K-, Ca-, and Mg-related traits simultaneously. This means that these traits might be controlled by the same QTL. In addition, we highlight that 4B might be an important chromosome regulating the nutrition of K, Ca, and Mg in wheat. The 4B chromosome and four hot QTL clusters, which located 45 QTLs, might be important potential targets for further investigation.

## 1. Introduction

Wheat (*Triticum aestivum* L.) contributes one-third of the world’s edible dry matter. It is one of the most important grain crops in the world. Potassium (K) is one of the essential macronutrients for crops, which is not only important for crop growth, development, and fecundity, but also significant for crop yield and quality [1]. It can increase the salt, drought, and disease tolerance of plants [2,3,4,5,6,7,8]. The average reserves of K in soil are usually large. However, most of the K in soil is not plant-available, and K deficiency is one of the most common limiting factors for crop production [9].

Potassium deficiency can significantly affect the use of K or other elements and ultimately affect crop yield [3,10]. There are complex interactions among K, calcium (Ca), and magnesium (Mg). Potassium can reduce the uptake of Mg in numerous plant species, such as soybeans, wheat, and rice [11,12,13]. However, the mechanisms for this K-inhibited Mg uptake have not been researched clearly. One possible explanation is the competition for apoplast binding between K^+^ and Mg^2+^ [14], while another possible explanation is the competition for the unidentified transporters between them. Tomoaki et al. [15] suggested that OsHKT2;4 (a K^+^-permeable transporter/channel)-mediated currents could also exhibit permeability to both Mg^2+^ and Ca^2+^, which would be smaller with the competitive inhibition of K^+^. Some genes or transporters/channels also showed sensitivity to Mg^2+^ and Ca^2+^ simultaneously [16], such as the *PaAlr1* gene in ascospores [17] and OsHKT2;4 in rice [15]. Ca can usually promote the uptake of K by plants but competition in absorption at the plasma membrane has also been observed [18]. In higher plants, some members of the CBL, CaM, and CML genes family of Ca^2+^ sensors have been reported to function in plant responses to K^+^ deficiency [19,20,21]. Xu et al. [21] showed that the CBL-CIPK (CBL-CIPK: CBL-interacting protein kinase) complex participates in the regulation of K^+^ uptake under K^+^ deficiency stress for plants. Obviously, a complex relationship in absorption or transport among K, Ca, and Mg exists widely, and these reports have provided us with some interpretation of it. However, to our knowledge, there have been few similar reports in wheat. Further genetic investigations should be carried out for K, Ca, and Mg nutrition in wheat.

The nutrient-related traits for K, Ca, and Mg are very complicated quantitative traits. Wheat is a very important crop that has a large genome. Until now, few genes related to plant nutrition were cloned in wheat, although the first high-affinity potassium uptake transporter, HKT1 in higher plants, was cloned in wheat [22]. Quantitative trait loci (QTL) analysis is still an effective way to identify the location of new genes, which dissects complicated traits into component loci and study their relative effects on a specific trait [23,24,25]. In wheat, QTL analysis has been used to study the effects of different nutrient environments [26,27,28,29], which enables us to understand nutrient use efficiency at the QTL level. However, to this day, few studies have been reported on verified QTLs for the efficient use of K, Ca, and Mg simultaneously in uniform environments. 

The main objective of our study was to identify the QTLs related to the absorption and utilization of K, Ca, and Mg at the seedling stage in a hydroponic culture trial and the mature stage in a field trial under different K treatments (LK and CK) using a set of RILs (recombinant inbred lines) derived from two winter wheat varieties of China. The results may help us further understand the effects of K deficiency on K, Ca, and Mg nutrition at the phenotypic and QTL level. They may also provide valuable QTLs for K, Ca, and Mg nutrition in wheat, which deserves further investigation.

## 2. Materials and Methods

### 2.1. Plant Materials

The RIL population (F9) used in this investigation was derived from a cross of “Tainong 18 × Linmai 6” using single-seed descent (SSD). A total of 184 lines were randomly selected from the original 305 lines of this population to construct the genetic map and QTL analysis [30]. The outstanding characteristics of Tainong 18 are high-yield, high quality, and resistance to lodging. It was planted in approximately 300 thousand hectares per year in the Huang-huai Winter Wheat Region of China. Linmai 6 is a high yield wheat variety belonging to the medium to large spike type, and its female parent is a sister line of the famous cultivars “Jimai 22”.

### 2.2. Experimental Design

#### 2.2.1. Hydroponic Culture Trial

The 184 RILs and their parents were grown in hydroponic culture in the greenhouse at Shandong Agricultural University in February 2013 and March 2013. Optimized Hoagland’s nutrient solution [31] (Appendix A) was used for the well-balanced growth of wheat seedling. Two treatments with moderate K (CK, 66.47 mg/L) and low K (LK, 6.65 mg/L) concentrations were designed with the consistent concentrations of other elements. A random complete block design was used in our experiments, with three replicates for each treatment.

A total of 50 seeds of each line and their parents were sterilized for five minutes in 10% H_2_O_2_. Then, they were germinated at 25 °C for seven days. Eighteen uniform seedlings (3 plants × 2 treatments × 3 replicates) with both the embryogenic primary root and coleoptile were selected and transferred to nutrient solution (20 L with one replication in lightproof container). The distances between various lines were 3 cm × 3 cm. The nutrient solutions were continuously aerated and renewed every 4 days. The 0.1 mmol∙L^−1^ HCl or NaOH solution was used to regulate the pH of nutrient solution between 6.0 and 6.2 every day. The plants grew for 28 days in nutrient solution and were harvested. The details referenced the method of Guo et al. [32].

#### 2.2.2. Field Trial

The field trials were carried out at the agronomy experimental station of Shandong Agricultural University from 2012–2013. The soil type was loamy cinnamon soil (pH 7.8). The average contents of N (available N), P (Olsen P), and K (available K) in the 0 to 20 cm soil profile sampled were 58.2, 21.3, and 86.4 mg·kg^−1^ without fertilizing. Two K concentration treatments, moderate K (114 kg/hm^2^) and low K (0 kg/hm^2^), were designed. In addition, 50% of the total N (97.5 kg/hm^2^), all the P_2_O_5_ (102 kg/hm^2^), and the corresponding K_2_O in two treatments were applied as base fertilizer before sowing, and the other 50% of N (97.5 kg/hm^2^) was applied at the stem elongation stage. Each treatment was replicated twice. Twenty seeds of each line were sown on October 10, and ten seedlings were retained after germination, with a 10 cm spacing between plants and 25 cm between rows. All of the materials were harvested on June 10. Twenty plants of each line in the same K treatment were put together as one sampling during harvest and then threshing for further testing.

### 2.3. Trait Measurements

All of the investigated traits and their abbreviations are listed in Table 1. For hydroponic trials, the three replicates for each line (nine plants) of each treatment were pooled together as one mixed sample and separated into roots and shoots. After being dried at 105 °C for two hours and dried at 60 °C for 72 h in an oven, the dry weight and the concentrations of K, Ca, and Mg in roots and shoots were measured. The concentrations of K, Ca, and Mg in roots and shoots were determined using atomic absorption spectroscopy (AA7000) after microwave digestion using HNO_3_. K-use efficiency (RKUE) was the ratio between dry weight and the concentration of K in the corresponding part of plant. For example, root K-use efficiency (RKUE) was the ratio between root dry weight per plant and the concentration of K in root (RDW/RKCE). The measurement methods of other nutrient use efficiencies were similar to the root K-use efficiency.

For field trials, the plant height (PH), spike number per plant (SN), and grain number per spike (GN) were determined from five random plants for each replicate of each line. All of the plants for each line of one K treatment were pooled together and then measured for dry weight and K, Ca, and Mg concentrations of the straw and grain, separately. The concentrations of K, Ca, and Mg in grains and straws were determined using atomic absorption spectroscopy (AA7000) after microwave digestion using HNO_3_. The calculation methods of nutrient use efficiencies were similar to the root K-use efficiency in the hydroponic culture trial. The K harvest index (KHI) is the ratio between grain K content per square meter and aboveground K content per square meter (GKC/AKC). The measurement methods of Ca and Mg harvest indexes were similar to the KHI.

### 2.4. Data Analysis

The SPSS 18.0 software (SPSS Inc., Chicago, IL, USA) was employed to conduct the analyses of variance (ANOVA), the least significant difference (LSD) test, and Spearman’s correlation coefficients (*r*) between different traits. In a no-repeat trial design, using a two-factor model was adequate for ANOVA. All factors including RILs (n − 1) degrees of freedom, treatments (t − 1), and random error ((n − 1)(t − 1)) were considered sources of random effects. Multiple comparison tests for the traits between “treatments” were calculated by taking all of the RILs as replicates and using the mean value of the same K condition for each trait. The variance of K conditions was excluded when the broad-sense heritability (h_B_^2^) was estimated according to the formula: h_B_^2^ = σg^2^/(σg^2^ + σe^2^), where *σg*^2^ was the genotypic variance and *σe*^2^ was the total error variance.

The high-density genetic map for 184 RILs of “TN18 × LM6” [28] was employed in the QTL analysis. The map comprised of 10739 loci (5399 unique loci) assigned to 21 chromosomes, with a total map length of 3394.47 cM and a density of 0.63 cM/marker. The Windows QTL Cartographer 2.5 software (Http://statgen.ncsu.edu/qtlcart/wqtlcart.htm) [33] was used to perform the QTL mapping in this study. The presence of the significant QTL was declared via the threshold that was defined by 1000 permutations at *p* ≤ 0.05 [34]. The identification of QTL cluster and its confidence interval referenced the results of the meta-analysis (Biomercator 2.0 software, AIC = 4 (model 4) in the step Meta-analysis 2/2 (http://www.genoplante.com)) [29].

### 2.5. Naming Method of QTLs

QTLs were named according to the method of “Q + trait name + chromosome name + experimental treatment.” Among them, traits are represented by their English abbreviation, and “−” was added between traits and chromosomes. QTLs for the same trait on the same chromosome were distinguished using an Arabic numeral (1, 2, 3, …). In addition, E1 and E2 stand for the hydroponic trial of February 2013 and March 2013, respectively. T1 and T2 stand for CK and LK treatments, respectively.

## 3. Results

### 3.1. Phenotypic Variation and Correlations Between Traits

The significant differences among most of the investigated traits of the RIL population were found in both the hydroponic and field trials (Table 2 and Table 3, Appendix A). The coefficients of variation (CVs; CV = SD/Average × 100%) exhibited wide ranges among the 184 RILs. They ranged from 9.51% to 50.43% in the hydroponic culture trial and from 6.63% to 51.27% in the field trials. The CVs for 69.50% of all the traits were greater than 20% in the two trials (Appendix A). The 31 and 38 investigated traits under hydroponic culture and field trials in each treatment (respectively) showed a continuous distribution. The h_B_^2^ for all investigated traits ranged from 50.16% (TKUE—total K-use efficiency) to 80.00% (RMgCE—concentration of root Mg) in the hydroponic trial and from 42.41% (StKCE—concentration of straw K) to 89.67% (TGW—thousand grains weight) in the field trials (Appendix A). The ANOVA results showed that the variance for either genotype or treatment effects on most investigated traits were significant at a *p* ≤ 0.05, excluding genotypic effects on SKCE (concentration of shoot K), TKCE (concentration of total K), SKUE (shoot K-use efficiency), and TKUE, as well as the treatment effects on RMgCE (concentration of root Mg) in the hydroponic culture trial. In the field trial, the genotypic effects on HI (harvest index), StKCE, AKCE (concentration of aboveground K), StMgCE (concentration of straw Mg), CaHI (Ca harvest index), MgHI (Mg harvest index), GKUE (grain K-use efficiency), GCaUE (grain Ca-use efficiency), and ACaUE (aboveground Ca-use efficiency), as well as the treatment effects on HI, GCaCE (concentration of grain Ca), StCaC (straw Ca content per plant), ACaC (aboveground Ca content per plant), MgHI, and GCaUE, were not significant at *p* ≤ 0.05 (Appendix A).

The LSD (least significant difference) test of the RIL population showed that the average values of the investigated traits were significantly different among the treatments of the hydroponic trial and the field trial in most cases (Appendix A). These results indicated that the treatments and genetic background assisted in explaining the overall phenotypic variation. Most correlation coefficients (*r*) among traits were significant at the *p* ≤ 0.01 level (Appendix A) in the hydroponic culture trial, except for the *r* between RCaC (root Ca content per plant) and RKC (root K content per plant), SKC (shoot K content per plant), TKC (Total K content per plant); TCaUE (total Ca-use efficiency) and RCaC, SCaC (shoot Ca content per plant), TCaC (total Ca content per plant). In the field trials, the correlation coefficients (*r*) between yield traits show that PH (plant height), SN (spike number per square meter), GWP (grain weight per plant), StWP (straw weight per plant), and AWP (total aboveground weight per plant) were significantly and positively correlated with each other. There were significant and negative correlations between TGW (thousand grains weight) and SN (spike number), HI (harvest index), and StWP (straw weight per plant) (Appendix A).

### 3.2. Major Characteristics of the QTLs in Different Trials

#### 3.2.1. Hydroponic Culture Trial

A total of 217 additive QTLs for 31 seedling traits were detected on 19 chromosomes, except for 2D and 7D (Figure 1, Appendix A). Of these, 28, 52, 67, and 70 QTLs were detected for four biomass weight traits (RDW, SDW, TDW, RSDW), K efficiency-related traits (RKCE, SKCE, TKCE, RKC, SKC, TKC, SKUE, TKUE, RKUE), Ca efficiency-related traits (RCaCE, SCaCE, TCaCE, RCaC, SCaC, TCaC, SCaUE, TCaUE, RCaUE), and Mg efficiency-related traits (RMgCE, SMgCE, TMgCE, RMgC, SMgC, TMgC, SMgUE, TMgUE, RMgUE), respectively. An individual QTL explained between 5.02 (RSDW—ratio of root and shoot dry weight) and 41.17% (RSDW) of the phenotypic variation, and the highest LOD value for a single QTL was 24.56 for the RSDW in the T1E2 treatment experiment (Appendix A). Three RHF-QTLs or relatively stable QTLs, which could be detected in at least two treatments (environments) for RMgC (Root Mg content per plant), RKC (Root K content per plant), and RDW (Root dry weight per plant), were located (Table 4). The average contributions of these RHF-QTLs ranged from 8.88% to 12.59%.

#### 3.2.2. Field Trial

A total of 89 additive QTLs for 33 adult traits (except StWP, AKCE, AMgCE, StMgC, AMgUE) were detected on 19 chromosomes, except for 2B and 3D (Figure 1, Appendix A). Of these, 21, 22, 35, and 11 QTLs were detected for yield traits (PH, SN, GN, TGW, GWP, AWP, HI), K efficiency-related traits (GKCE, StKCE, GKC, StKC, AKC, KHI, GKUE, StKUE, AKUE), Ca efficiency-related traits (GCaCE, StCaCE, ACaCE, GCaC, StCaC, ACaC, CaHI, GCaUE, StCaUE, ACaUE), and Mg efficiency-related traits (GMgCE, StMgCE, GMgC, AMgC, MgHI, GMgUE, StMgUE), respectively. An individual QTL explained between 7.04% (CaHI—Ca harvest index) and 21.66% (GKUE—grain K-use efficiency) of the phenotypic variation, and the highest LOD value for a single QTL was 10.53 for the GKUE in the CK treatment (Appendix A). One RHF-QTLs or relatively stable QTLs for TGW (thousand grains weight) were located, with contributions of 8.08% (Table 4).

### 3.3. QTL Clusters

A total of 18 QTL clusters (C1–C18) (a cluster was defined as the co-location of QTLs for more than two traits) were mapped to nine chromosomes, involving 96 out of the 306 QTLs (31.37%) (Figure 1, Table 5). All these QTL clusters could be classified into three types: detected only for seedling traits (Type I, including C1, C2, C4, C6–C14, and C16), only for adult traits (Type II, including C15 and C18), and simultaneously for seedling and adult traits (Type III, including C3, C5, and C17). Of these QTL clusters, C7, C12, C16, and C18 (Table 5, Figure 1) were the most important for seedling traits and/or adult traits.

## 4. Discussion

### 4.1. K Effects on Biomass, K-, Ca-, and Mg-Related Traits of the RIL Population

Potassium is one of the essential nutrient elements for wheat. A deficiency in K can slow plant growth and decrease biomass production [35,36]. Compared with the CK treatment, the seedling traits of SDW and TDW significantly decreased in the LK treatment. In a similar manner, the SH, SN, GN, TGW, GWP, StWP, and AWP of maturity traits all decreased with reduced K concentration (Appendix A).

It has been widely reported that there is a competitive relationship among K, Ca, and Mg as it relates to absorption [13,20,21,37]. The uptake of K may be affected by Mg, while Mg is affected by K [15]. Similarly, uptake of Ca could also be depressed by increasing the concentration of Mg [38]. In this study, the K content-related traits decreased, and the K use efficiency-related traits increased with the decreasing K supply compared to the normal treatment at the seedling and mature stage. In contrast, content-related traits of Ca and Mg increased, while the use efficiency-related traits decreased in the LK treatment at the seedling and maturity stage. These results also indicated that there was an antagonistic effect in absorption among K, Ca, and Mg.

### 4.2. K Effects on QTLs for K, Ca, and Mg Efficiency-Related Traits

For wheat, nutrient treatment can significantly affect the expression of nutrient-related QTLs. Some studies of QTLs or QTL clusters had been conducted under different nitrogen concentrations [27,39,40,41,42]; under conditions of different K concentrations [28]; and under various concentrations of N, P, and K treatments [32]. In different nutrient environments, the number and the location of most QTLs detected for certain traits were quite different in these previous experiments. Similar to these studies, 302 (98.69%) QTLs and 10 QTL clusters (including 55 QTLs) of this study were detected only once in a single (moderate K or low K) K treatment (Appendix A, Table 5). These sites might be important for adaptation to different K environments.

This indicated that K treatment greatly affected the expression of QTLs related to K, Ca, and Mg, and there might be quite different mechanisms for K, Ca, and Mg nutrition under different K treatments. The QTL clusters also provided some evidence for this indication.

### 4.3. Hot QTL Clusters for K, Ca, and Mg Efficiency-Related Traits

Many studies have reported that there were some genes which can affect the absorption of K, Ca, or Mg simultaneously. For example, the CBL-CIPK (CBL-CIPK: CBL-interacting protein kinase, Ca^2+^ sensors) complex participates in the regulation of plant K^+^ uptake under K^+^-deficiency stress [21]. The OsHKT2;4 (a K^+^-permeable transporter/channel)-mediated currents could also exhibit permeability to both Mg^2+^ and Ca^2+^ [15]. Finally, some genes or transporters/channels, such as the *PaAlr1* gene in ascospore [17] and OsHKT2;4 in rice [15] showed sensitivity to Mg^2+^ and Ca^2+^ simultaneously [16]. The QTL clusters in this investigation might also provide us with some evidence about that. Eight QTL clusters included QTLs for K, Ca, and Mg simultaneously under the same K treatment, and these clusters included 61 QTLs. C7, C12, and C16 involved QTLs for K, Ca, and Mg simultaneously (C7, C16 in CK; C12 in LK); C5, C9, and C10 involved QTLs for K and Ca simultaneously (C10 in LK; C5, C9 in CK); and C15 and C18 (in LK) involved QTLs for K and Mg simultaneously. These QTL clusters highlight the important hot sites on the chromosome where some genes might be located that can affect the absorption of K, Ca, or Mg simultaneously for wheat and required further investigation.

### 4.4. The Importance of the 4B Chromosome

Our study highlights the importance of chromosome 4B. We detected 68 QTLs (22.22%) and 10 QTL clusters (55.56%; C3–C12) on the chromosome 4B−1 involving biomass traits and K, Ca, and Mg efficiency-related traits (Appendix A, Table 5). The four RHF-QTLs in this paper were also on chromosome 4B−1. The average contributions of QTLs in C5, C6, C7, C8, and C12 were all greater than 10%. We found three important locations on the chromosome 4B: D−1098436—D−1068778, S-3941408—D−1138250, and Tdurum_contig37811_134—wsnp_RFL_Contig3236_3262140. On the D-3570086—D−1098436, we found one RHF-QTL (in C3): QTGW.1-4B (T1, T2). On the S-3941408—D−1138250 and Tdurum_contig37811_134—wsnp_RFL_Contig3236_3262140, we found two QTL clusters (C7 and C12). They included QTLs for biomass, K, Ca, and Mg simultaneously. In addition, we found three RHF-QTLs (QRDW-4B (T1E2, T2E1), QRKC-4B (T1E2, T2E1), QRMgC.1-4B (T1E2, T2E1), in C12) on Tdurum_contig37811_134—wsnp_RFL_Contig3236_3262140. The contributions of these three RHF-QTLs were 10.32%, 11.56%, and 17.42%, respectively. These three locations may include important genes that can regulate the nutritional properties of K, Ca, and Mg.

Recently, Yuan et al. [29] reported the QTL mapping for P efficiency and morphological traits in wheat used the same RIL population derived from a cross of “Tainong 18 × Linmai 6.” They found that four (C3, C4, C5, C6) out of 10 clusters were mapped to the 4B−1 chromosome and that the C3 and C5 clusters contained one and four RHF-QTLs, respectively. It is worth noting that many similar locations of QTL clusters were found between Yuan et al. [29] and this study (Table 6). These results showed that 4B is a very important chromosome for mineral nutrition related to P, K, Ca, and Mg in wheat. In addition, the chromosome 4B also contained two (C3 and C5) out of three QTL clusters that contained QTLs, both for seedling and adult traits. The chromosome 4B and the QTL clusters are obvious and important targets that require further investigation.

## Figures and Tables

**Figure 1 genes-10-00607-f001:**
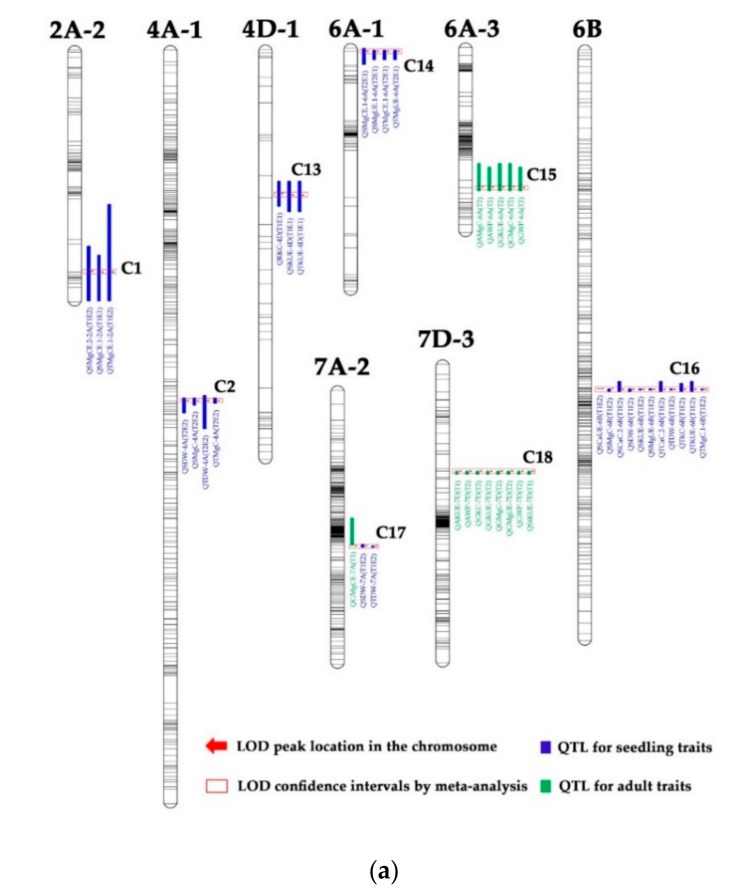
Locations of QTL clusters for wheat seedling and adult traits in different K treatment based on RILs derived from Tainong18 × Linmai6. In the names of the QTLs, E1 and E2 stand for the hydroponic trials of February 2013 and March 2013, respectively. T1 and T2 stand for the CK and LK treatments, respectively.

**Table 1 genes-10-00607-t001:** Summary of investigated traits and their abbreviations.

Abbr.	Meaning	Units	Abbr.	Meaning	Units
Seedling Trait			Adult Trait		
RDW	Root dry weight per plant	mg·plant^−1^	PH	Plant height	cm
SDW	Shoot dry weight per plant	mg·plant^−1^	SN	Spike number	No.
RSDW	Ratio of root and shoot dry weight		GN	Grain number per spike	No.
TDW	Total dry weight per plant	mg·plant^−1^	TGW	Thousand grains weight	g
RKC	Root K content per plant	mg·plant^−1^	GWP	Grain weight per plant	g/plant
SKC	Shoot K content per plant	mg·plant^−1^	StWP	Straw weight per plant	g/plant
TKC	Total K content per plant	mg·plant^−1^	AWP	Total aboveground weight per plant	g/plant
RCaC	Root Ca content per plant	mg·plant^−1^	HI	Harvest index	
SCaC	Shoot Ca content per plant	mg·plant^−1^	GKC	Grain K content per plant	mg/plant
TCaC	Total Ca content per plant	mg·plant^−1^	StKC	Straw K content per plant	mg/plant
RMgC	Root Mg content per plant	mg·plant^−1^	AKC	Aboveground K content per plant	mg/plant
SMgC	Shoot Mg content per plant	mg·plant^−1^	GCaC	Grain Ca content per plant	mg/plant
TMgC	Total Mg content per plant	mg·plant^−1^	StCaC	Straw Ca content per plant	mg/plant
RKCE	Concentration of root K	g/kg	ACaC	Aboveground Ca content per plant	mg/plant
SKCE	Concentration of shoot K	g/kg	GMgC	Grain Mg content per plant	mg/plant
TKCE	Concentration of total K	g/kg	StMgC	Straw Mg content per plant	mg/plant
RCaCE	Concentration of root Ca	g/kg	AMgC	Aboveground Mg content per plant	mg/plant
SCaCE	Concentration of shoot Ca	g/kg	GKCE	Concentration of grain K	g/kg
TCaCE	Concentration of total Ca	g/kg	StKCE	Concentration of straw K	g/kg
RMgCE	Concentration of root Mg	g/kg	AKCE	Concentration of aboveground K	g/kg
SMgCE	Concentration of shoot Mg	g/kg	GCaCE	Concentration of grain Ca	g/kg
TMgCE	Concentration of total Mg	g/kg	StCaCE	Concentration of straw Ca	g/kg
RKUE	Root K-use efficiency	mg/(µg·mg^−1^)	ACaCE	Concentration of aboveground Ca	g/kg
SKUE	Shoot K-use efficiency	mg/(µg·mg^−1^)	GMgCE	Concentration of grain Mg	g/kg
TKUE	Total K-use efficiency	mg/(µg·mg^−1^)	StMgCE	Concentration of straw Mg	g/kg
RCaUE	Root Ca-use efficiency	mg/(µg·mg^−1^)	AMgCE	Concentration of aboveground Mg	g/kg
SCaUE	Shoot Ca-use efficiency	mg/(µg·mg^−1^)	GKUE	Grain K-use efficiency	g/(mg·g^−1^)
TCaUE	Total Ca-use efficiency	mg/(µg·mg^−1^)	StKUE	Straw K-use efficiency	g/(mg·g^−1^)
RMgUE	Root Mg-use efficiency	mg/(µg·mg^−1^)	AKUE	Aboveground K-use efficiency	g/(mg·g^−1^)
SMgUE	Shoot Mg-use efficiency	mg/(µg·mg^−1^)	GCaUE	Grain Ca-use efficiency	g/(mg·g^−1^)
TMgUE	Total Mg-use efficiency	mg/(µg·mg^−1^)	StCaUE	Straw Ca-use efficiency	g/(mg·g^−1^)
			ACaUE	Aboveground Ca-use efficiency	g/(mg·g^−1^)
			GMgUE	Grain Mg-use efficiency	g/(mg·g^−1^)
			StMgUE	Straw Mg-use efficiency	g/(mg·g^−1^)
			AMgUE	Aboveground Mg-use efficiency	g/(mg·g^−1^)
			KHI	K harvest index	
			CaHI	Ca harvest index	
			MgHI	Mg harvest index	

**Table 2 genes-10-00607-t002:** Average values for the RILs for each treatment and trait of the hydroponic trial.

Treatment	Traits ^e^	Average ^f^	Traits ^e^	Average ^f^	Traits ^e^	Average ^f^
CKE1	RKCE	34.05a	RCaCE	0.99b	RMgCE	0.60a
LKE1	g/kg	15.99b	g/kg	1.53a	g/kg	0.59a
CKE2		26.95a		1.45b		0.77a
LKE2		5.10b		2.06a		0.78a
CKE1	SKCE	43.72a	SCaCE	4.60b	SMgCE	2.39b
LKE1	g/kg	21.42b	g/kg	9.32a	g/kg	2.98a
CKE2		43.66a		7.38b		2.53b
LKE2		12.27b		10.32a		3.79a
CKE1	TKCE	41.98a	TCaCE	3.94b	TMgCE	2.06b
LKE1	g/kg	20.11b	g/kg	7.43a	g/kg	2.41a
CKE2		39.92a		6.04b		2.13b
LKE2		10.41b		8.18a		3.02a
CKE1	RKC	0.67a	RCaC	18.79b	RMgC	11.83a
LKE1	mg/plant	0.30b	µg/plant	27.91a	µg/plant	11.09a
CKE2		0.58a		30.62b		16.46a
LKE2		0.10b		38.08a		14.80b
CKE1	SKC	3.90a	SCaC	407.64b	SMgC	211.81a
LKE1	mg/plant	1.25b	µg/plant	543.87a	µg/plant	174.41b
CKE2		3.20a		535.38b		183.47b
LKE2		0.67b		556.82a		205.70a
CKE1	TKC	4.57a	TCaC	426.44b	TMgC	223.64a
LKE1	mg/plant	1.55b	µg/plant	571.11a	µg/plant	185.59b
CKE2		3.78a		566.00b		199.93b
LKE2		0.77b		594.85a		220.53a
CKE1	RKUE	0.58b	RCaUE	23.12a	RMgUE	36.01a
LKE1	mg/(µg∙mg^−1^)	1.17a	mg/(µg∙mg^−1^)	15.40b	mg/(µg∙mg^−1^)	36.31b
CKE2		0.79b		16.40a		27.97a
LKE2		3.73a		10.01b		24.40b
CKE1	SKUE	2.07b	SCaUE	20.54a	SMgUE	38.84a
LKE1	mg/(µg∙mg^−1^)	2.76a	mg/(µg∙mg^−1^)	6.51b	mg/(µg∙mg^−1^)	20.12b
CKE2		1.70b		10.30a		29.56a
LKE2		4.56a		5.43b		14.48b
CKE1	TKUE	2.63b	TCaUE	29.14a	TMgUE	54.57a
LKE1	mg/(µg∙mg^−1^)	3.86a	mg/(µg∙mg^−1^)	10.75b	mg/(µg∙mg^−1^)	32.80b
CKE2		2.40b		16.21a		45.14a
LKE2		7.19a		9.22b		24.52b

^e^ RKCE: concentration of root K; SKCE: concentration of shoot K; TKCE: concentration of total K; RKC: root K content per plant; SKC: shoot K content per plant; TKC: total K content per plant; RKUE: root K-use efficiency; SKUE: shoot K-use efficiency; TKUE: total K-use efficiency; RCaCE: concentration of root Ca; SCaCE: concentration of shoot Ca; TCaCE: concentration of total Ca; RCaC: root Ca content per plant; SCaC: shoot Ca content per plant; TCaC: total Ca content per plant; RCaUE: root Ca-use efficiency; SCaUE: shoot Ca-use efficiency; TCaUE: total Ca-use efficiency; RMgCE: concentration of root Mg; SMgCE: concentration of shoot Mg; TMgCE: concentration of total Mg; RMgC: root Mg content per plant; SMgC: shoot Mg content per plant; TMgC: total Mg content per plant; RMgUE: root Mg-use efficiency; SMgUE: shoot Mg-use efficiency; TMgUE: total Mg-use efficiency. ^f^ Within the same column followed by the same letter means the associated values did not differ significantly according to the LSD test (*p* < 0.05).

**Table 3 genes-10-00607-t003:** Average values for the RILs for each treatment and trait of field trial.

Treatment	Traits ^e^	Average ^f^	Traits ^e^	Average ^f^	Traits ^e^	Average ^f^
CK	GKCE	3.78a	GCaCE	0.40a	GMgCE	1.46b
LK	g/kg	3.10b	g/kg	0.41a	g/kg	1.49a
CK	StKCE	21.17a	StCaCE	3.68b	StMgCE	1.54b
LK	g/kg	16.03b	g/kg	4.39a	g/kg	1.60a
CK	AKCE	12.84a	ACaCE	2.11b	AMgCE	1.50b
LK	g/kg	9.81b	g/kg	2.48a	g/kg	1.54a
CK	GKC	73.00a	GCaC	7.73a	GMgC	28.12a
LK	mg/plant	53.66b	mg/plant	7.12b	mg/plant	25.78b
CK	StKC	450.24a	StCaC	78.48a	StMgC	32.76a
LK	mg/plant	304.83b	mg/plant	82.62a	mg/plant	29.96b
CK	AKC	523.68a	ACaC	86.20a	AMgC	60.84a
LK	mg/plant	358.44b	mg/plant	89.71a	mg/plant	55.72b
CK	KHI	0.14b	CaHI	0.09a	MgHI	0.47a
LK		0.16a		0.08b		0.46a
CK	GKUE	5.20b	GCaUE	51.49a	GMgUE	13.26a
LK	g/(mg·g^−1^)	5.62a	g/(mg·g^−1^)	48.53b	g/(mg·g^−1^)	11.69b
CK	StKUE	1.03b	StCaUE	5.95a	StMgUE	14.07a
LK	g/(mg·g^−1^)	1.21a	g/(mg·g^−1^)	4.46b	g/(mg·g^−1^)	12.08b
CK	AKUE	3.22b	ACaUE	19.71a	AMgUE	27.19a
LK	g/(mg·g^−1^)	3.67a	g/(mg·g^−1^)	15.28b	g/(mg·g^−1^)	23.62b

^e^ GKCE: concentration of grain K; StKCE: concentration of straw K; AKCE: concentration of aboveground K; GKC: grain K content per plant; StKC: straw K content per plant; AKC: aboveground K content per plant; KHI: K harvest index; GKUE: grain K-use efficiency; StKUE: straw K-use efficiency; AKUE: aboveground K-use efficiency; GCaCE: concentration of grain Ca; StCaCE: concentration of straw Ca; ACaCE: concentration of aboveground Ca; GCaC: grain Ca content per plant; StCaC: straw Ca content per plant; ACaC: aboveground Ca content per plant; CaHI: Ca harvest index; GCaUE: grain Ca-use efficiency; StCaUE: straw Ca-use efficiency; ACaUE: aboveground Ca-use efficiency; GMgCE: concentration of grain Mg; StMgCE: concentration of straw Mg; AMgCE: concentration of aboveground Mg; GMgC: grain Mg content per plant; StMgC: straw Mg content per plant; AMgC: aboveground Mg content per plant; MgH: Mg harvest index; GMgUE: grain Mg-use efficiency; StMgUE: straw Mg-use efficiency; AMgUE: aboveground Mg-use efficiency. ^f^ Within the same column followed by the same letter means the associated values did not differ significantly according to the LSD test (*p* < 0.05).

**Table 4 genes-10-00607-t004:** RHF-QTLs detected in more than two treatments under the hydroponic culture trial and field trial.

Traits	QTLs	Treatment	Marker Intervals	Additive Effects			Contributions (%)		
				Min	Max	Average	Min	Max	Average
Hydroponic culture trial									
RMgC	QRMgC.1-4B (T1E2 T2E1)	CK, LK	Tdurum_contig37811_134 wsnp_RFL_Contig3236_3262140	1.21	1.25	1.23	7.44	10.32	8.88
RKC	QRKC-4B (T1E2 T2E1)	CK, LK	Tdurum_contig37811_134 wsnp_RFL_Contig3236_3262140	0.03	0.05	0.04	9.43	11.56	10.50
RDW	QRDW-4B (T1E2 T2E1)	CK, LK	Tdurum_contig37811_134 wsnp_RFL_Contig3236_3262140	1.53	2.44	1.98	7.76	17.42	12.59
Field trial									
TGW	QTGW.1-4B(TI T2)	CK, LK	D−1098436-D−1103601	0.93	1.09	1.01	7.18	8.98	8.08

RMgC: root Mg content per plant; RKC: root K content per plant; RDW: root dry weight per plant; TGW: thousand grains weight.

**Table 5 genes-10-00607-t005:** Clusters comprising QTLs for more than two traits at seedling and mature stage.

Type	Code	Chromosome	Marker Intervals	No. of QTLs	QTLs for Seedling Traits			QTLs for Adult Traits	
**I**	C1	2A-2	wPt−1480-D-4008129	3	QSMgCE.1-2A (T1E1)	QSMgCE.2-2A (T1E2)	QTMgCE.1-2A (T1E2)		
	C2	4A−1	S-3957023-S−1073520	4	QTDW-4A (T2E2)	QSDW-4A (T2E2)	QSMgC-4A (T2E2)		
					QTMgC-4A (T2E2)				
	C4	4B−1	D−111318-Ku_c63300_1309	3	QSDW.1-4B (T1E1)	QTDW.1-4B (T1E1)	QSKUE.1-4B (T1E1)		
	C6	4B−1	D-3940950-D−1673295	4	QSMgUE.1-4B (T1E1)	QSMgCE-4B (T1E1)	QTMgCE.1-4B (T1E1)		
					QRSDW.3-4B (T2E2)				
	C7	4B−1	S-3941408-D−1138250	15	QSKUE.2-4B (T1E1)	QTKUE-4B (T1E1)	QSKC.2-4B (T1E2)		
					QSCaUE.1-4B (T1E1)	QSKC.1-4B (T1E1)	QTCaUE.1-4B (T1E1)		
					QTKC.2-4B (T1E1)	QSCaCE.2-4B (T1E2)	QSDW.2-4B (T1E1)		
					QTDW.2-4B (T1E1)	QTMgUE.1-4B (T1E1)	QRSDW.1-4B (T1E1)		
					QTCaCE.2-4B (T1E2)	QSCaCE.1-4B (T1E1)	QSKCE-4B (T1E2)		
	C8	4B−1	D-4008856-D-3943712	3	QRSDW.2-4B (T1E2)	QSCaUE.3-4B (T1E2)	QSCaC.2-4B (T2E2)		
	C9	4B−1	D−1302339-D-3024409	4	QSCaCE.3-4B (T1E2)	QTCaC.2-4B (T2E2)	QSDW.3-4B (T1E1)		
					QSKUE.3-4B (T1E1)				
	C10	4B−1	D−1380792-D−1094306	4	QRCaCE.2-4B (T2E2)	QSCaC.3-4B (T2E2)	QSKC.4-4B (T2E2)		
					QTKC.4-4B (T2E2)				
	C11	4B−1	wmc657-D−1666781	4	QRCaC.2-4B (T1E2)	QRMgC.2-4B (T2E2)	QSCaC.1-4B (T1E2)		
					QTCaC.1-4B (T1E2)				
	C12	4B−1	Tdurum_contig37811_134-	11	QRMgC.1-4B (T1E2 T2E1)	QRKC-4B (T1E2 T2E1)	QRDW-4B (T1E2 T2E1)		
			wsnp _RFL_Contig3236_3262140		QSDW.4-4B (T2E1)	QTDW.3-4B (T2E1)	QTKC.3-4B (T2E1)		
					QSMgUE.3-4B (T2E1)	QSKC.3-4B (T2E1)	QTMgUE.2-4B (T2E1)		
					QTCaUE.3-4B (T2E1)	QSCaUE.4-4B (T2E1)			
	C13	4D−1	wPt-732418-wmc825	3	QRKC-4D (T1E1)	QSKUE-4D (T1E1)	QTKUE-4D (T1E1)		
	C14	6A−1	S−1378596-D-3952397	4	QSMgCE.1-6A (T2E1)	QSMgUE.1-6A (T2E1)	QTMgCE.1-6A (T2E1)		
					QTMgUE-6A (T2E1)				
	C16	6B	D-3384656-D-3960242	11	QSCaUE-6B (T1E2)	QSMgC-6B (T1E2)	QSCaC.2-6B (T1E2)		
					QTCaC.2-6B (T1E2)	QTKUE-6B (T1E2)	QTKC-6B (T1E2)		
					QSDW-6B (T1E2)	QSKUE-6B (T1E2)	QSMgUE-6B (T1E2)		
					QTDW-6B (T1E2)	QTMgC.1-6B (T1E2)			
**II**	C15	6A-3	D-4329389-S−1091880	5				QAMgC-6A (T2)	QAWP-6A (T2)
								QGKUE-6A (T2)	QGMgC-6A (T2)
								QGWP-6A (T2)	
	C18	7D-3	D-3033829-D−1668160	8				QAKUE-7D (T1)	QAWP-7D (T2)
								QGKC-7D (T2)	QGKUE-7D (T2)
								QGMgC-7D (T2)	QGMgUE-7D (T2)
								QGWP-7D (T2)	QStKUE-7D (T1)
**III**	C3	4B−1	D−1098436-D−1068778	4	QSCaCE.4-4B (T2E2)	QTCaCE.3-4B (T2E2)		QTGW.1-4B (TI T2)	
					QTCaUE.2-4B (T1E2)				
	C5	4B−1	Ku_c63300_1309-D-3022151	3	QTCaCE.1-4B (T1E1)			QTGW.2-4B (T2)	QGKUE-4B (T1)
	C17	4B−1	D-3025056-D-3956324	3	QSDW-7A (T1E2)	QTDW-7A (T1E2)		QGMgCE-7A (T1)	

SMgCE: concentration of shoot Mg; TMgCE: concentration of total Mg; TDW: total dry weight per plant; SDW: shoot dry weight per plant; SMgC: shoot Mg content per plant; TMgC: total Mg content per plant; SKUE: shoot K-use efficiency; SMgUE: shoot Mg-use efficiency; RSDW: ratio of root and shoot dry weight; TKUE: total K-use efficiency; SKC: shoot K content per plant; SCaUE: shoot Ca-use efficiency; TCaUE: total Ca-use efficiency; TKC: total K content per plant; TMgUE: total Mg-use efficiency; TCaCE: concentration of total Ca; SCaCE: concentration of shoot Ca; SKCE: concentration of shoot K; SCaC: shoot Ca content per plant; TCaC: total Ca content per plant; RCaCE: concentration of root Ca; RCaC: root Ca content per plant; RMgC: root Mg content per plant; RKC: root K content per plant; RDW: root dry weight per plant; SMgUE: shoot Mg-use efficiency; AMgC: aboveground Mg content per plant; AWP: total aboveground weight per plant; GKUE: grain K-use efficiency; GMgC: grain Mg content per plant; GWP: grain weight per plant; AKUE: aboveground K-use efficiency; GKC: grain K content per plant; GMgUE: grain Mg-use efficiency; StKUE: straw K-use efficiency; TGW: thousand grains weight; GMgCE: concentration of grain Mg.

**Table 6 genes-10-00607-t006:** QTL clusters detected in the same or adjacent marker regions in this paper and in Yuan et al. [29].

Code/Type	Chromosomes	Marker Intervals	No. of QTLs	QTLs	
C3/II [29]	4B−1	D−1051883-D−1113185	3	QSdw.1	QTpute.1
				QGpute.1	
C4/III	4B−1	D−1113185-Ku_c63300_1309	3	QSDW.1-4B (T1E1)	QTDW.1-4B (T1E1)
				QSKUE.1-4B (T1E1)	
C4/II [29]	4B−1	D-3022151-D−1040960	5	QRspc.1	QSn.2
				QSl	QGn.1
				QRpc.2	
C5/III	4B−1	Ku_c63300_1309-S−1040960	4	QTCaCE.1-4B (T1E1)	QTGW.2-4B (T2)
				QGKUE-4B (T1)	
C5/II [29]	4B−1	D−1083795-D-3940950	8	QSdw.2	QTdw
				QRsdw	QSpute.2
				QTpute.2	QSn.3
				QGwp	QGpute.2
C6/I	4B−1	D-3940950-D−1673295	4	QSMgUE.1-4B (T1E1)	QSMgCE-4B (T1E1)
				QTMgCE.1-4B (T1E1)	QRSDW.3-4B (T2E2)
C6/II [29]	4B−1	D-4008856-D−1138250	4	QGn.2	QRspc.2
				QSdw.3	QPh
C7/I	4B−1	S-3941408 - D−1138250	15	QSKUE.2-4B (T1E1)	QTKUE-4B (T1E1)
				QSKC.2-4B (T1E2)	QSCaUE.1-4B (T1E1)
				QSKC.1-4B (T1E1)	QTCaUE.1-4B (T1E1)
				QTKC.2-4B (T1E1)	QSCaCE.2-4B (T1E2)
				QSDW.2-4B (T1E1)	QTDW.2-4B (T1E1)
				QTMgUE.1-4B (T1E1)	QRSDW.1-4B (T1E1)
				QTCaCE.2-4B (T1E2)	QSCaCE.1-4B (T1E1)
				QSKCE.-4B (T1E2)	

Tpute: total P-utilization efficiency; Gpute: grain P-utilization efficiency; Rspc: ratio of root and shoot content; Sl: spike length; Rpc: root P-content per plant; Spute: shoot P-utilization efficiency; Ph: plant height; SDW: shoot dry weight per plant; TDW: total dry weight per plant; SKUE: shoot K-use efficiency; TCaCE: concentration of total Ca; TGW: thousand grains weight; GKUE: grain K-use efficiency; SMgUE: shoot Mg-use efficiency; SMgCE: concentration of shoot Mg; TMgCE: concentration of total Mg; RSDW: ratio of root and shoot dry weight; TKUE: total K-use efficiency; SKC: shoot K content per plant; SCaUE: shoot Ca-use efficiency; TCaUE: total Ca-use efficiency; TKC: total K content per plant; SCaCE: concentration of shoot Ca; TMgUE: total Mg-use efficiency; SKCE: concentration of shoot K.

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
