# Peer review of "The Hot QTL Locations for Potassium, Calcium, and Magnesium Nutrition and Agronomic Traits at Seedling and Maturity Stages of Wheat under Different Potassium Treatments"

_genes, 2019, doi:10.3390/genes10080607_

Round 1
Reviewer 1 Report
“The Hot QTL Locations for Potassium, Calcium, Magnesium Nutrition and Morphological Traits at Seedling and Maturity Stages of Wheat under Different Potassium Treatments “ by Shen et al.
With the aim to understand the impact of K+ starvation on nutrient use efficiency in the wheat, Shen et al., identified quantitative trait loci (QTLs) for K+, Ca2+ and Mg2+. Two stages of wheat development, seedling and maturity, are studied. The authors observed that under reduced K+ supply, plant K+ content decreased and K+ use efficiency related traits increased, whereas the opposite was found for Ca2+ and Mg2+. They also identified several QTL clusters controlling all the studied cations related traits simultaneously. Moreover, their results pinpointed that 4B-chromosome is important for mineral nutrition and contains major QTLs related to K+, Ca2+ and Mg2+ efficiency.
The study by Shen et al represents a lot of work and the experiments are well conducted.
It is inconvenient that some paragraphs are very similar to different places in the text such as lines 267-272 and 45-49 and elsewhere.
Please check that K+, Ca2+ and Mg2+ are correctly written.
Reviewer 2 Report
The paper is devoted to an important subject of nutrient use inheritance in wheat. The paper is scientifically solid and presents good results. However, the presentation of the finding can be improved substantially.
The main challenge is too many traits and abbreviations so that it is difficult to follow. Many traits are highly related. It would be useful to identify the key traits and focus on them making more sense to biological interpretations. The bi-plot analysis would be very useful to present the traits relationship. It would be useful to present the traits absolute values in the paper and not in supplement. Especially the average values for the RILs for each treatment and trait shall be presented. The ultimate objective of the analysis is to help select the best lines. Are there RIL lines superior to parents for the nutritional traits? If yes, - they can be presented with the main traits and key QTLs in a supplement. Sections 4.2, 4.3 and 4.4 of the discussion look like results with reference to tables and figures. Have to me modified. In the paper title "morphological" can be replaced with "agronomic". Line 75: RIL lines - which generation? Line 78: Panicle type of variety - what does it mean? Line 98: mistake in the title.
